# A Statistical Approach to Assess Biological Equivalent Mean Organ Dose (MOD) for Different Fractionations in Thoracic Radiotherapy

**DOI:** 10.3390/biomedicines13051110

**Published:** 2025-05-02

**Authors:** Siyao Zhong, Guangshan Wang, Xiaohang Qin, Yinghui Li, Bin Sun, Feng-Ming (Spring) Kong, Yuyan Gao, Jian-Yue Jin

**Affiliations:** 1Department of Radiotherapy, Beijing Luhe Hospital, Capital Medical University, Beijing 101100, China; 19910620569@163.com (S.Z.); 18611537249@163.com (B.S.); 2Department of Radiotherapy, Beijing Chest Hospital, Capital Medical University, Beijing 101100, China; wang_guangshan@163.com; 3School of Biomedical Engineering, Capital Medical University, Beijing 100071, China; mooneve_211@163.com (X.Q.); 15858114919@163.com (Y.L.); 4Department of Clinical Oncology, Hong Kong University Shen-Zhen Hospital, Shenzhen 518053, China; kong0001@hku.hk

**Keywords:** lung cancer, radiotherapy, biological equivalent dose, mean organ dose, organ-at-risk

## Abstract

**Background:** With advances in radiotherapy technology, there are more technological options and fractionation regiments for different stages of lung cancer. Evaluation of the possibility of severe organ toxicity for the organ-at-risks (OARs) is often required when treating patients with unusual fractionations or combining two treatment plans with different fractionations. **Purpose:** This study aims to provide an approach that can reasonably estimate the possibility of toxicity using a biological equivalent mean organ dose (MOD)-based model from the 2 Gy per fraction era for various fractionations. **Methods and Materials:** The dose volume histograms (DVHs) of 272 patients treated with radiotherapy for lung cancer at a prescribed dose of 2 Gy × 30 f were used for analysis. For each patient, we calculated the biological equivalent MOD based on a dose conversion of EQD2 (equivalent to a dose of 2 Gy/fraction to the organ), the MOD based on the physical dose, and the d-value that makes the biological equivalent MOD based on a dose conversion of EQDd equal to the physical MOD. Statistical analysis was then performed to determine the relationship of the d-value to the corresponding physical MOD in percentage dose (relative MOD). **Results:** Equivalent MODs based on EQD2 were smaller than the physical MOD for each patient, suggesting that using EQD2 conversion would underestimate the equivalent MOD. The distribution of d-values with physical relative mean lung dose (RMLD) showed a normal peak with an average of d = 1.3 Gy, suggesting that the equivalent MLD can be estimated using EQD1.3 for various fractionations. Regression analysis showed that the d-value could be estimated as d = $\sqrt{0.93+3.0\times RMLD}$, d = $\sqrt{0.044+5.8\times RMED}$, and d = $\sqrt{4.7\times RMHD}$ for the lung, esophagus, and heart, respectively, suggesting that the equivalent MLD, MED, and MHD can be estimated using EQDd. **Conclusions:** Using EQD2 conversion underestimates the equivalent MOD, and the equivalent MOD converted by EQDd with an appropriate d-value may enhance the assessment of organ toxicity.

## 1. Background

Lung cancer remains the leading cause of cancer mortality worldwide [1,2]. The treatment of lung cancer is complex, often involving multiple treatment modalities. Radiotherapy (RT) plays an important role, with about 77% of lung cancer patients having an evidence-based indication for radiotherapy at some point in their cancer journey [3,4].

With advances in RT technology, there are more technological options as well as fractionation regiments for different stages of lung cancer. The standard prescription is usually 60 Gy in 30 fractions for stage III non-small-cell lung cancer (NSCLC) [5,6,7]; however, two-phase adaptive RT with varying fractionations according to the shrinking target volume in the middle treatment simulation images has also been proposed [8]. Stereotactic ablative body radiotherapy (SABR) with 8–20 Gy per fraction in 3–5 fractions is often used for patients with medically inoperable stage I–IIA NSCLC [9,10]. Some recent studies have shown that hypofractionated Intensity-Modulated RT with 50 to 80 Gy in 3 to 4 Gy fractions provides favorable rates of local control and survival for some well-selected patients [11,12,13]. The standard treatment for stage I–III small-cell lung cancer (SCLC) is twice-daily treatment with 45 Gy in 30 fractions, while some centers use daily treatment with 45 Gy in 15 fractions. Different centers also use several other fractionations [14,15]. In addition, patients with a primary lung tumor are at high risk of locoregional recurrence, a second primary lung cancer or thoracic metastases. Patients with other primary tumors may also experience metastasis to the lungs. These patients may require repeated treatment and benefit from re-irradiation [16,17,18].

Evaluation of possible organ toxicity for the organs-at-risk (OARs) is required when treating patients with these unusual fractionations or combining two treatment plans with different fractionations [19,20,21]. The lungs, esophagus, and heart are the primary OARs for lung cancer [22,23,24,25]. Their mean organ dose (MOD) is an important parameter to evaluate the probability of toxicities. However, the same total physical dose with different fractionations can produce distinct biological effects. Therefore, a MOD-based toxicity model from the conventional 2-Gy per fraction era may not be applied to these different fractionations. A conversion from the physical dose to the equivalent dose at 2-Gy per fraction (EQD2) is often currently used to estimate the MOD [26,27,28,29]. The EQD2 conversion is fine for the target volume because it has a relatively uniform dose distribution of 2 Gy per fraction in the entire target volume. However, as described in the result section, using EQD2 usually underestimates the equivalent MOD for the OARs [16,30,31], because the majority of the OAR volumes have doses much less than 2 Gy per fraction. Therefore, EQDd, with d < 2 Gy should be used to convert the physical MOD to the equivalent MOD. The purpose of this research aims to determine the d-values for the lung, heart, and esophagus so that a reasonable equivalent MOD corresponding to the 2-Gy per fraction era can be estimated.

## 2. Methods and Materials

### 2.1. Basic Approach

We first define EQDd as the equivalent dose in *d* Gy per fraction to the organ or a normalization of the biological equivalent dose (BED) with a fractionation of *d* Gy per fraction. Our proposed dose conversion approach is based on the finding that for each patient with a conventional dose prescription of 2 Gy × 30 fractions, there is a dose *d* that the equivalent MOD based on a dose conversion with EQDd equals to the MOD based on the physical dose. Given a dose volume histogram (DVH) with BED, we can calculate the MOD corresponding to the 2 Gy per fraction era by normalizing the BED into the EQDd. Therefore, the question becomes how to determine the d-value. We used a large number of DVHs from real patients with a 2 Gy × 30 prescription to calculate the d for each patient and finally determined the d expression.

### 2.2. DVHs of Real Patients

DVH data of 272 lung cancer patients treated with a standard prescribed dose (2 Gy × 30 f for PTV) were retrospectively collected from several hospitals. All plans were calculated using similar constraints for the OARs. In this paper, we are primarily concerned with the mean dose of the most common OARs for lung cancer, the mean lungs dose (MLD), the mean esophagus dose (MED), and the mean heart dose (MHD). In addition, the Vn of the lung (L-V_n_), esophagus (E-V_n_), and heart (H-V_n_) were also evaluated.

### 2.3. BED Calculation

DVH data of the lung, esophagus, and heart with absolute volume (V_i_) vs. absolute physical dose (D_i_) were exported from treatment plans. V_i_ represents the irradiated volume of the derived structure at dose point D_i_; the value of D_i_ ranges from 0 Gy to the maximum dose point D_max_ at a dose interval of 0.1 Gy. The simple LQ model-based BED equation without corrections for tumor proliferation, time, or other biological factors was used for the derivation. It is expressed as(1)$$BED=D_{i}(1+d_{i}/(\alpha /\beta ))$$

And can be normalized as EQDd as(2)$${\left(EQDd\right)}_{i}=\frac{D_{i}(1+d_{i}/(\alpha /\beta ))}{1+d/(\alpha /\beta )}$$
where d_i_ is the fraction size of D_i_, and the α/β values for the target volume were 10, and for the OARs (lungs, esophagus, and heart), they were 3.

### 2.4. Derivation of d for Each Patient

We first calculated the MOD for the original DVH with a physical dose using the equation(3)$$MOD=\sum_{i=1}\frac{V_{i}-V_{i-1}}{V}\times D_{i}$$
where V is the total volume of the calculated OAR.

We then used EQDd to replace the physical dose in the DVH with d = 0.1, 0.2, … 2 Gy and calculated MOD_EQDd_ using the equation(4)$${MOD}_{EQDd}=\sum_{i=1}\left(\frac{V_{i}-V_{i-1}}{V}\times {\left(EQDd\right)}_{i}\right)$$

Using an iteration method, we derived the d-value that satisfies MOD = MOD_EQDd_. We derived the d-value for each of the 272 cases for the lung, esophagus, and heart, respectively, and plotted the d-value with the corresponding MOD in percentage dose.

### 2.5. Statistical Analysis

Descriptive statistics such as the median, maximum, and minimum were calculated for the mean dose of OARs and the corresponding d-value. Correlation and regression analyses were used to test the correlation and degree of correlation between d-values and mean dose. ORIGIN version 18 was used for statistical analysis.

## 3. Results

### 3.1. Evaluation of the Mean Dose

Figure 1 shows the DVHs of the PTV (Figure 1a), lung (Figure 1b), esophagus (Figure 1c), and heart (Figure 1d) for a representative patient to compare the MOD_EQDd_ of various d with the MOD of the original physical dose. The DVH for the d-value that satisfies MOD_EQDd_ = MOD was plotted. In addition, the DVH of EQDd with d-values of 0.5, 2 (most commonly used value), and 2.5 Gy were also plotted for comparison. For the PTV, the DVH curve of the EQD2 after conversion for each physical dose point is almost identical to the original curve (Figure 1a), because a majority of the PTV received a prescription dose of 2 Gy × 30 f. On the other hand, the EQD2 dose distribution for the three OARs is lower than the physical dose distribution. Consequently, the MLD, MED, and MHD based on EQD2 were 12.2%, 9.2%, and 12.7% lower than that based on the physical dose, respectively. A lower than 2 Gy d-value has to be used for the EQDd to achieve MOD_EQDd_ = MOD for the three OARs.

Table 1 lists the mean dose for the PTV, lung, esophagus, and heart for various d-values were listed in Table 1. The d-values that satisfy MOD_EQDd_ = MOD were 2.0, 1.4, 1.5, and 1.4 Gy for the PTV, lungs, esophagus, and heart, respectively.

### 3.2. Normality Test for d-Value

All lungs, esophagus, and heart d-value data on 272 patients were analyzed using a normality test. The test results are shown in Figure 2. The d-value for lungs ranged from 0.8 Gy to 1.8 Gy, with a mean of 1.33 Gy, a standard error of 0.2 Gy, and a median of 1.3 Gy, and the statistics are consistent with a normal distribution (Figure 2a), suggesting that an average d-value of 1.3 Gy may be used for the majority of patients. The d-values for the esophagus ranged from 0.5 Gy to 2 Gy, with a mean of 1.55 Gy, a standard error of 0.33 Gy, and a median of 1.6 Gy, and the statistics did not conform to a normal distribution (Figure 2b). The d-values for the heart ranged from 0.1 Gy to 1.8 Gy, with a mean of 0.97 Gy, a standard error of 0.46 Gy, and a median of 1.1 Gy, and the statistics did not conform to a normal distribution (Figure 2c).

### 3.3. Derivation of d-Value Expression Using Regression

We found that the d-value correlated to the relative mean dose to the lungs (RMLD = MLD/Prescribed dose), esophagus (RMED), and heart (RMHD).

As can be seen from the scatter plot in Figure 3a, the distribution of the RMLD and corresponding d-value is relatively concentrated, the range for the RMLD is 10–50% and the d-value is between 0.8 and 1.8 Gy. The results of the F-test of the nonlinear regression analysis showed that the RMLD affects the d-value (F = 6619, *p* < 0.001). The R-squared is 0.137, indicating that the RMLD determines only 13.7% of the corresponding d-value. The regression coefficient is 3.0, which indicates that the RMLD positively affects the d-value. The regression equation is(5)$$d=\sqrt{0.93+3.0\times RMLD}$$

The RMED is highly dependent on the distance between the esophagus and the target, so the RMED and its corresponding d-value have a wide range of distributions; the range of the RMED is 0–80%, and the d-value is between 0.5 and 2 Gy (Figure 3b). The same is true for the RMHD; the range of the RMHD is about 0–80%, and the d-value is between 0.1 and 1.8 Gy (Figure 3c).

The results of the F-test of the nonlinear regression analysis showed that the RMED affects the d-value (F = 8972, *p* < 0.001), The R-squared value is 0.666, indicating that the RMED determines 66.6% of the corresponding d-value. The regression coefficient is 5.8. It indicates that the RMED positively affects the d-value. The regression equation is(6)$$d=\sqrt{0.044+5.8\times RMED}$$

The results of the F-test of the linear regression analysis showed that the RMHD affects the d-value (F = 3171, *p* < 0.001). The R-squared value is 0.775, indicating that the RMHD determines 77.5% of the corresponding d-value. The regression coefficient is 4.7, which indicates that the RMHD positively affects the d-value. The regression equation is(7)$$d=\sqrt{4.7\times RMHD}.$$

### 3.4. Application for Re-Irradiation

A lung cancer patient with receiving re-irradiation was used as an example to illustrate the application of our approach. The prescribed dose for the first course (Course1) of radiotherapy is 3 Gy × 20 f, and the prescribed dose for re-irradiation (Course2) is 4 Gy × 8 f. The DVHs for the PTV, lungs, esophagus, and heart were derived. The total mean dose can be directly summed after BED conversion. As shown in Table 2, the total physical MLD, MED, and MHD are 17.8, 29.5, and 29.9 Gy, respectively. The biological equivalent total MLD, MED, and MHD based on EQD2 are 16.8, 28.0, and 27.8 Gy, respectively, less than the corresponding physical mead doses. Considering that the dose per fraction for both courses is larger than 2 Gy, theoretically, the biological mean doses should be larger than the physical mean doses. Therefore, the biological equivalent mean doses based on EQD2 are evidently unreasonable. Using our regression equations, the corresponding d-values for the MLD, MED, and MHD in Course1 are 1.3, 1.6, and 1.4 Gy, respectively, and 1.1, 0.8, and 1 Gy, correspondingly, for course 2. Consequently, the total MOD based on EQDd conversion is 19.7, 31.0, and 32.1 Gy, respectively, higher than the physical mean dose. The total MLD, MED, and MHD increased by 10.7%, 10.5%, and 10.7%, respectively.

## 4. Discussion

This study firstly pointed out that using the EQD2 to calculate the biological equivalent MOD is incorrect and would underestimate the biological equivalent MOD and, consequently, the probability of toxicities. Instead, using the EQDd with a d-value less than 2 Gy would be a better approach. We performed a statistical analysis of DVH data from 272 lung cancer patients who received radiotherapy with a prescribed dose of 2 Gy × 30 fractions to determine the best d-value. The analysis indicates that the d-value varies with different patients and different organs. The d-values for the lung have a relatively narrow normal distribution, suggesting that an average d-value of 1.3 Gy can be used to convert the BED of the lungs to the EQD1.3 to evaluate the MLD in patients with unconventional fractionation.

In addition, we found that d-values correlated to the relative mean organ dose (RMOD), regression analyses result indicates that the RMLD determines 13.7% of the corresponding d-value for the lung, and the d-value can be calculated by d = $\sqrt{0.93+3.0\times RMLD}$; the RMED determines 66.6% of the corresponding d-value for the esophagus, and the d-value can be calculate by d = $\sqrt{0.044+5.8\times RMED}$; the RMHD determines 77.5% of the corresponding d-value for heart, and the d-value of MED can be calculate by d = $\sqrt{4.7\times RMHD}$.

Thoracic radiotherapy with unusual fractionations, such as SBRT and hypo-fractionated radiotherapy, often uses the BED for dose assessment [32,33]. However, the dose limit criteria with BED for OARs are unclear, and there are no OAR toxicity models based on BED. Most toxicity models are based on the OAR doses in the 2 Gy per fraction data. Some studies convert the BED to EQD2 to evaluate the plan relative to conventional fractionation [20,21,34]. However, as corroborated earlier, the conversion of OAR doses using EQD2 consistently underestimates of the MOD. This study provides an approach to estimate the biologically equivalent MOD for a more accurate prediction of toxic responses using toxicity models based on the MOD in 2 Gy per fraction data. The approach can also be applied to evaluate the toxicities for re-irradiated patients, while previous dosimetric studies of re-irradiation could only evaluate the MOD separately for the two plans [35,36].

The d-value appears to vary relatively small for the lung but relatively large for the esophagus and heart. The frequency histogram of the d-values for the lungs shows a Gaussian distribution, while they are apparently not Gaussian for the esophagus and heart. Because most of the lung cancers have target volumes that are either adjacent to or within the lung volumes, the dose to the lungs tends to be relatively consistent across patients, leading to a relatively narrow distribution of d-value for the lungs. This narrow distribution also indicates that the average d-value of 1.3 Gy can also be used to evaluate the equivalent MLD. In contrast, the average doses for the esophagus and heart are more variable, as they are influenced by their distance from the target volume. This variability in average doses across patients leads to a more dispersed distribution of d-values, and Equations (6) and (7) have to be used to estimate d-value for the equivalent MED and MHD.

This study has some limitations. It only provides methods for evaluating the MOD of different fractionations for thoracic radiotherapy, and the calculation of the MOD is based on statistical analysis of data from multiple patients. The d-value varies with different patients. Although we used the RMOD to estimate the best d-value for each patient, the predicted d-value may contain uncertainties. In addition, irradiated percentage volume at a dose of n Gy (V_n_) is an important parameter for evaluating the lungs, esophagus, and heart, and this paper does not examine how to assess V_n_ for different fractionations.

## 5. Conclusions

This study provides an approach to assess the potential toxicities for the lungs, esophagus, and heart in thoracic radiotherapy using MOD-based toxicity models from the 2-Gy per fraction era for different fractionations. Our results indicate that using the EQD2 conversion would underestimate the biological equivalent MOD. A conversion using the EQDd, with the d-value calculated as d = $\sqrt{0.93+3.0\times RMLD}$, d = $\sqrt{0.044+5.8\times RMED},$ and d = $\sqrt{4.7\times RMHD}$, respectively, for the lungs, esophagus, and heart, may improve the assessment. This study has clinical significance because there are no dosimetry-based toxicity models for unconventional dose fractionations.

## Figures and Tables

**Figure 1 biomedicines-13-01110-f001:**
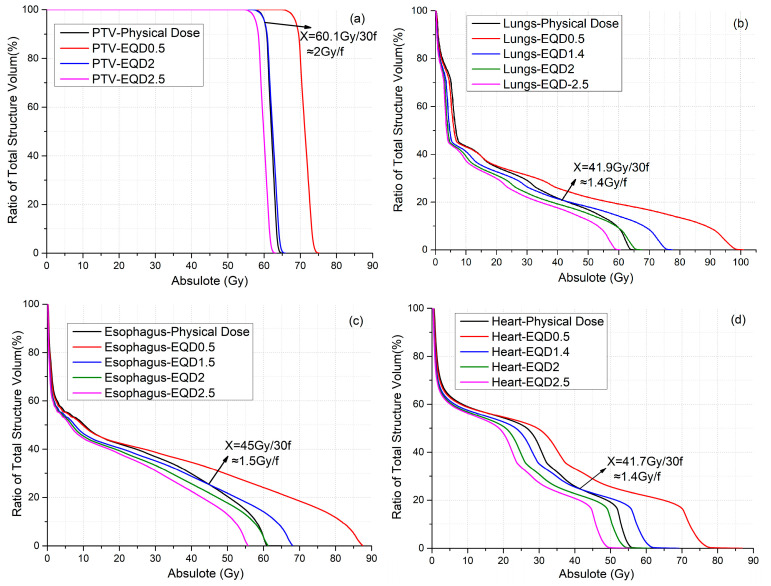
(**a**) DVH with EQD0.5, EQD2, and EQD2.5 and physical dose for PTV. (**b**) DVH with EQD0.5, EQD1.4, EQD2, and EQD2.5 and physical dose for lungs. (**c**) DVH with EQD0.5, EQD1.5, EQD2, and EQD2.5 and physical dose for esophagus. (**d**) DVH with EQD0.5, EQD1.4, EQD2, and EQD2.5 and physical dose for heart.

**Figure 2 biomedicines-13-01110-f002:**
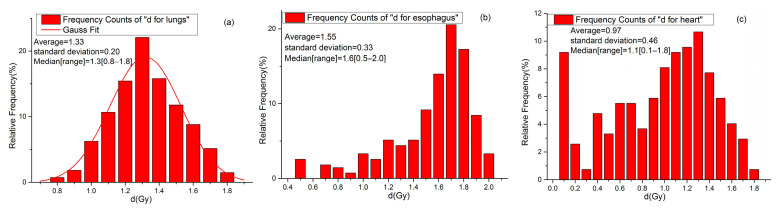
(**a**) Frequency histogram of d-values for lungs. (**b**) Frequency histogram of d-values for esophagus. (**c**) Frequency histogram of d-values for heart.

**Figure 3 biomedicines-13-01110-f003:**
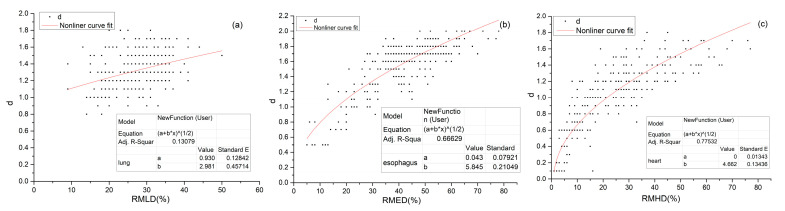
Scatter plot and regression equation for the d-value corresponding to different (**a**) RMLD and the d-value, (**b**) the RMED and (**c**) the RMHD. Note: The * in the figure indicates multiplication (“×”).

**Table 1 biomedicines-13-01110-t001:** PTV and OAR mean dose at different EQD values.

Mean Dose	Physical Dose	EQD0.5	EQDd	EQD2	EQD2.5
PTV	61.9 Gy	71.2 Gy	62.3 Gy (d = 2)	62.3 Gy	59.8 Gy
MLD	19.6 Gy	25.7 Gy	19.8 Gy (d = 1.4)	17.2 Gy	15.4 Gy
MED	21.7 Gy	28.1 Gy	21.9 Gy (d = 1.5)	19.7 Gy	17.9 Gy
MHD	23.6 Gy	29.4 Gy	23.4 Gy (d = 1.4)	20.6 Gy	18.7 Gy

**Table 2 biomedicines-13-01110-t002:** Mean dose of EQDd and EQD2 for each structure.

Structure		Course1	Course2	Total
MLD (Gy)	Physical dose	14.73 (RMLD = 0.2455)	3.08 (RMLD = 0.096)	17.8
	EQDd	15.72 (d = 1.3)	3.99 (d = 1.1)	19.7
	EQD2	13.51	3.27	16.8
MED (Gy)	Physical dose	25.91 (RMHD = 0.4318)	3.56 (RMHD = 0.1113)	29.5
	EQDd	27.27 (d = 1.6)	3.77 (d = 0.8)	31.0
	EQD2	25.09	2.86	28.0
MHD (Gy)	Physical dose	24.57 (RMED = 0.4095)	5.36 (RMED = 0.1675)	29.9
	EQDd	25.99 (d = 1.4)	6.11 (d = 1)	32.1
	EQD2	22.87	4.88	27.8

## Data Availability

The datasets used and/or analyzed during the present study are available from the corresponding author on reasonable request.

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
