# Peer review of "A Statistical Approach to Assess Biological Equivalent Mean Organ Dose (MOD) for Different Fractionations in Thoracic Radiotherapy"

_biomedicines, 2025, doi:10.3390/biomedicines13051110_

Round 1
Reviewer 1 Report
Comments and Suggestions for Authors
- The capitalization of keywords should be consistent.
- The background introduction of "Radiation oncology teams often face the issue of organ-at-risk (OARs)" is insufficient and should include additional relevant content.
- The abbreviation for "Figure" should be consistent in “Figure 1 shows the DVHs of the PTV (Fig. 1a), lung (Fig. 1b), esophagus (Fig. 1c)…”
- Tab 2 in line 208 should also be verified.
Further polishing of English seems to be better
Reviewer 2 Report
Comments and Suggestions for Authors
Please provide the importance of study in your conclusion part
Factors Influencing the Determination of MOD Using EQDd in Patient-Specific Treatment Plans?
How do changes in physical dose affect the mean dose evaluation for OARs?
How does the RMLD influence the d-value in lung radiation therapy?
Can the proposed approach for estimating biologically equivalent MOD improve patient outcomes in re-irradiation cases?
Please expand the discussion part with more recent and relevant references
Comments on the Quality of English LanguageQuality of English can be enhanced by improving sentence delivery in more scientific way
Reviewer 3 Report
Comments and Suggestions for Authors
In the current statistically based retrospective study, Siyao Zong and colleagues from Capital Medical University (PRC) attempted to optimize treatment regimens for lung cancer patients receiving radiotherapy, aiming to find mean organ doses for lung, esophagus and heart. By introducing the term d-value, defining the optimized Gy/fraction for each organ, using the iteration method, and replacing the physical dose in the dose-volume histogram with the equivalent dose of d Gy/fraction (EQDd) followed by normality test, the authors found that EQD2, widely used in clinical practice for lung, should be replaced by EQD1.3/fraction to reduce the toxic load while maintaining the efficacy of the effect. In addition, the authors calculated d for esophagus and heart, which were also less than 2 (underestimation of MOD), and performed similar calculations for the re-irradiation scheme. The results obtained by the authors, although not experimentally validated, are of high value for the optimization of radiotherapy regimens for patients and will be of interest to a wide range of readers of Biomedicines. Before accepting this paper for publication, the authors should make corrections according to the following remarks: (1) Dear authors, please give your suggestions as to why the frequency histogram of d-values for lungs fits the Gaussian distribution while the other models do not? (2) Line 111 - Please give your explanation why alpha/beta for the OAR for lung, esophagus and heart was taken as 3? (3) A number of corrections: Line 44 - Please sort out the labeling of the references. Here they are in parentheses and in line 52 they are in square brackets. Line 53 and throughout the manuscript - please add space between the value and its unit (for istance, 8-20 Gy). Line 86 - please change x to × Please rearrange Table 1 and Figure 1 (position as mentioned in the text). Line 154 - please add space between figure and 2. Line 169 - please add space between figure and 3(a). Lines 179, 180 - please add space in "Gy(Fig".
